# Identification and Analysis of Corrosion Mechanisms for Ground Pipelines with Hanging Rings

**Yuntao Xi** [1,2], **Yidi Li** [1], **Yang Yao** [3,4], **Qingming Gan** [3,4], **Yixu Wang** [5], **Lei Wang** [1,6,7,8,*], **Lei Wen** [7], **Shilei Li** [6], **Daoyong Yang** [2,*], **Jiangtao Ji** [9] and **Shubin Lei** [1]

1 School of Material Science and Engineering, Xi'an Shiyou University, Xi'an 710065, China
2 Petroleum Systems Engineering, Faculty of Engineering and Applied Science, University of Regina, Regina, SK S4S 0A2, Canada
3 Oil&Gas Technology Research Institute of Changqing Oilfield Company, PetroChina, Xi'an 710018, China
4 National Engineering Laboratory for Exploration & Development of Low Permeability Oil and Gas Fields, Xi'an 710018, China
5 Metropolitan College, Boston University, Boston, MA 02215, USA
6 State Key Lab of Advanced Metals and Materials, University of Science and Technology Beijing, Beijing 100083, China
7 Nation Center for Materials Service Safety, University of Science and Technology Beijing, Beijing 100083, China
8 State Key Laboratory of Metastable Materials Science and Technology, Yanshan University, Qinhuangdao 066004, China
9 China Railway First Survey and Design Institute Group Co., Ltd., Xi'an 710043, China
* Correspondence: wanglei@xsyu.edu.cn (L.W.); tony.yang@uregina.ca (D.Y.); Tel.: +86-157-7192-1910 (L.W.)

**Abstract:** Recently, corrosion perforation has been frequently seen in surface pipelines in the oil and gas industry, resulting in operational and environmental challenges. Due to the complex characteristics and mechanisms of such corrosion, a new and pragmatic method has been designed to identify and evaluate the corrosion phenomenon via a hanging ring installed in a surface pipeline. In addition to respectively analyzing the ions of water samples with chemical titration, ion chromatography, and mass spectrometry, the micro-surface morphology of the corroded hanging rings was observed and evaluated by using a scanning electron microscope (SEM) equipped with energy dispersive spectroscopy (EDS), and the surface composition of the corroded hanging rings was analyzed by using X-ray diffraction (XRD). The water ions of each selected position were found to mainly contain $Ca^{2+}$, $Ba^{2+}$, $SO_4^{2-}$, and $HCO_3^-$, while the barium scale and calcium carbonate scale were formed in situ. In addition to the common corrosion induced by $CO_2$, corrosion induced by both $CO_2$ and $H_2S$ leads to extremely serious corrosion and scaling in surface pipelines. In addition, the injection dose of corrosion inhibitor was also evaluated.

**Keywords:** surface pipeline; corrosion perforation; hanging ring; corrosion scaling

## 1. Introduction

To transport oil and natural gas resources, pipelines are one of the safest and most economical modes [1–3] and are commonly made of carbon steel because of its low price and excellent mechanical properties [4]. The produced water of the oil field dissolves various salts and gases in formation in the high-temperature and high-pressure oil layer. The fluid flows out from the oil and gas layers and flows through the wellbore and wellhead to the surface system. Due to changes in temperature, pressure and oil, and the gas and water balance, corrosive gases, such as $H_2S$ and $CO_2$, as well as dissolved oxygen introduced at the back end of the pipeline, and under the comprehensive action of high salinity water, pipelines can become severely corroded. The dissolved salts in oilfield water have a significant impact on the corrosion of water. When the dissolved salts are very low, the corrosion degree of different anions and cations to water is also different. Chloride ions,

carbonate and bicarbonate are common dissolved salts in oilfield water. In general, the corrosivity of water containing dissolved salts increases with the increase of dissolved salt concentration, and tends to decrease after the maximum value appears. Not only is carbon steel generally weak in regards to corrosion resistance, but it is also prone to internal and external corrosion during oil and gas transportation. The existence of carbon dioxide ($CO_2$) and hydrogen sulfide ($H_2S$) aggravates the corrosion on carbon steel pipelines [5,6] since, in an oil–water environment, $CO_2$ reacts with the produced water to form carbonic acid ($H_2CO_3$) [7] and $H_2S$ is dissolved in the produced water to induce $H^+$ ionization together with a series of corrosion reactions [8], respectively.

It is well known that the corrosion on carbon steel in an environment where $CO_2$ and $H_2S$ coexist is different to that of either pure $CO_2$ or pure $H_2S$ environments [9–11]. Pure $CO_2$ corrosion generally not only shows a uniform and local corrosion morphology, but also generates a layer of corrosion products on the surface of carbon steel [12,13]. Pure $H_2S$ corrosion usually shows local corrosion morphology, forms corrosion products under wetting conditions, and, finally, induces pitting or pits on the surface of carbon steel [14,15]. For carbon steel, $H_2S$ can accelerate anodic reactions (dissolution of carbon steel) [16] and the corrosion rate of steel can be increased by five times, due to the existence of $H_2S$ in a corrosive environment [17]. Silva et al. [18] identified the degradation mechanisms driven by hydrogen embrittlement in a mixed $CO_2/H_2S$ environment which has been experimentally verified. Zeng et al. [19] also experimentally found that hydrogen embrittlement dictates the failure of a sacrificial anode protector in a mixed $CO_2/H_2S$ environment. By evaluating and comparing the corrosion behavior of an S phase and AISI 304 austenitic stainless steel in a $H_2S/CO_2/Cl^-$ medium, Li et al. [20] found that the synergistic corrosion effect of $CO_2$ and $H_2S$ can be essentially attributed to the $CO_2$ enhanced $H_2S$ corrosion. When $CO_2$ and $H_2S$ are present at the same time, the corrosion product formed on the surface of carbon steel is a mixture of iron carbonate and iron sulfide, which play a certain role in retarding the corrosion and prevent further dissolution inside the surface corrosion product [21]. When the structure of iron carbonate and iron sulfide is not compact and the surface is loose and porous, such a corrosion product will not have its retarding function but, rather, accelerates the corrosion of carbon steel instead [22,23]. Previous studies on $CO_2$ and $H_2S$ corrosion were conducted in indoor environments. No attempts have been made to monitor and evaluate the corrosion phenomenon in a field case. Although such a study can reflect the corrosion and scaling of the transmission pipeline in the field environment to a certain extent, it cannot accurately examine the comprehensive effects of various factors. At the same time, the corrosion inhibition rate of the corrosion inhibitor is affected by such factors as inhibitor type, temperature, inhibitor concentration, time, flow rate, injection method, surfactant, gas-liquid ratio, etc. Therefore, many influencing factors and their interactions should be considered when using a corrosion inhibitor. The common corrosion inhibitors are divided into organic corrosion inhibitors and inorganic corrosion inhibitors. The application of corrosion inhibitors should consider the amounts of their addition and corresponding electrochemical performances.

In this study, customized hanging rings were used to evaluate and analyze the corrosion behavior in surface pipelines of the Jiyuan pool. Meanwhile, the main corrosion factors were identified. By performing water quality analysis, as well as SEM and XRD analyses, the underlying corrosion mechanisms of the surface pipelines were analyzed and identified. In addition, several common corrosion inhibitors were evaluated electrochemically.

## 2. Field Background

The Jiyuan pool, located in the mid-west of the Ordos Basin, spanning from Dingbian County (Shaanxi Province) to Yanchi County (Ningxia Hui Autonomous Region), is one of the main oil production regions in the Changqing Oilfield in China. After more than ten years of exploration and development, the Jiyuan pool has entered its stable production period, with increased water cuts and frequent leaks on the surface pipelines, due to corrosion, seriously restricting safe production, increasing maintenance costs, and threatening

the safety of the on-site operational staff [24,25]. For example, there are 45 corroded oil wells (including 20 severely corroded wells) in the whole area. Most of the produced water mainly contains calcium chloride with its total salinity as high as $4–11 \times 10^4$ mg/L and the $Cl^-$ concentration up to $1–7 \times 10^4$ mg/L, while corrosion and scaling are commonly encountered in the gathering and transmission systems. From 2018 to 2021, a total of 85 corrosion and leakage sections were replaced with a total length of 142 km, and internal corrosion perforation was the main failure mode.

## 3. Experimental

### 3.1. Materials

The hanging rings used in the experiment were made of 20# steel, as shown in Figure 1. The chemical composition was determined by spark spectrometer, as shown in Table 1. Prior to each test, the hanging ring was polished with 400#, 800#, 1000#, and 2000# sandpaper step by step, and then cleaned with acetone and alcohol to remove the oil. Once this was completed, it was weighed on an analytical balance with an accuracy of 0.01 mg, and the sample number and corresponding mass were recorded. Each polished hanging ring was well sealed with a sealing bag prior to its installment. Then, the hanging rings were placed at different stations along the pipeline, and were taken out after 3 months for corrosion and scaling analyses. In this way, the corrosion conditions at different positions of the pipeline could be monitored.

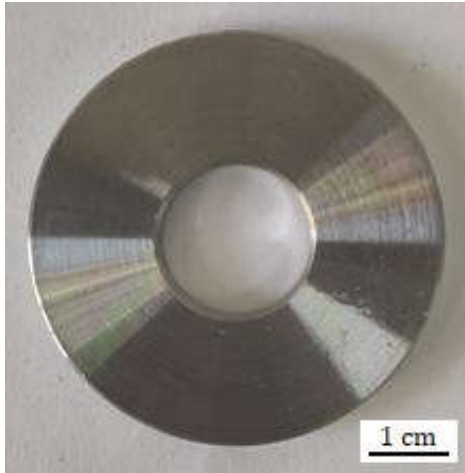

**Figure 1.** A customized hanging ring.

**Table 1.** Chemical composition of experimental hanging ring (wt.%).

| C | Si | Mn | P | S | Cr | Mo | Ni | Cu | B | Co | Nb | Ti | V | Fe |
|------|------|------|------|------|------|--------|------|------|--------|-------|--------|--------|--------|-------|
| 0.20 | 0.22 | 0.49 | 0.02 | 0.02 | 0.02 | <0.01 | 0.03 | 0.04 | <0.001 | 0.006 | <0.001 | <0.001 | <0.005 | 98.94 |

### 3.2. Water Quality Analysis of Produced Liquids

For pipelines with frequent corrosion and leakage, on-site water sampling was carried out. The water samples were contained in sealed 500 mL sample bottles (the sample bottles were sterilized before sampling), and taken to the indoor laboratory for full analysis and water quality testing with chemical titration, ion chromatography, and mass spectrometry. The water samples of 9 different positions with serious corrosion, which may have been caused by water ions, in the Jiyuan pool were collected and numbered from 1 to 9.

### 3.3. Characterization of Corrosion Products

Firstly, the morphology of a hanging ring used in the experiment was characterized by macro-photography to understand its corrosion characteristics. Then, a scanning electron

microscope (SEM) was used to analyze its micro-morphology, and point scanning and micro area surface scanning of energy dispersive spectroscopy (EDS) was used to further analyze the elemental composition of the surface corrosion/rust layer or the scaling layer. X-ray diffraction (XRD) was used to analyze the chemical composition of corrosion products and scale samples. The specific test parameters included the Cu target, acceleration voltage 40 kV, current 30 mA, and diffraction angle (2θ) of 10°–80°. The measured data were analyzed by using the Jade 6 phase analysis software.

### 3.4. Evaluation of Corrosion Inhibitors

In order to slow down the corrosion rate, the electrochemical performance of five common corrosion inhibitors (HHS08-1#, HHS08-3#, PD43-1, RX-21, and YD-03) was evaluated, and the best corrosion inhibitor and the best injection scheme were evaluated and determined. In the electrochemical tests, the base material was 20# steel, and the filling concentration of each inhibitor was 50, 80, 100, and 120 ppm, respectively. The simulated corrosion medium was prepared with the ion concentration of the site with serious corrosion, and the electrochemical test was carried out at room temperature and atmospheric pressure.

## 4. Results

### 4.1. Water Ion Analysis

As for the nine collected water samples, their ion analysis results are tabulated in Table 2. As can be seen, the ion content from every collected site in the Jiyuan pool was generally too high and the water fell into the following two main categories: calcium chloride and sodium bicarbonate. The concentrations of $Cl^-$, $HCO_3^-$, $SO_4^{2-}$, $Ca^{2+}$, $Mg^{2+}$, and $Ba^{2+}$ were 7465–51,795 mg/L, 215–1064 mg/L, 31–1807 mg/L, 0–435 mg/L, 0–223 mg/L, and 170–1486 mg/L, respectively. The water was mainly the calcium chloride type and sodium bicarbonate type. As for Sample #2, the $Cl^-$ content even reached as high as 47,050.94 mg/L, which was far beyond the normal level. This might be significantly related to the corrosion perforation of ground pipelines [17]. At the same time, the water ion of each sample had abundant $Ca^{2+}$, $Ba^{2+}$, $SO_4^{2-}$, and $HCO_3^-$, and it was easy to form barium scale and calcium carbonate scale in situ.

**Table 2.** Analysis results of water ion composition in different sample positions (mg/L).

| Sample Position | $Ba^{2+}$ | $Ca^{2+}$ | $Na^+ + K^+$ | $SO_4^{2-}$ | $HCO_3^-$ | $Cl^-$ | Water Type |
|---|---|---|---|---|---|---|---|
| 1 | 707.36 | 64.12 | 21,612.92 | 31.28 | 617.43 | 34,093.68 | Sodium bicarbonate |
| 2 | 1019.06 | 268.50 | 30,669.44 | 285.30 | 791.37 | 47,050.94 | Sodium bicarbonate |
| 3 | 278.35 | 40.79 | 20,021.06 | 862.16 | 1064.8 | 29,829.90 | Sodium bicarbonate |
| 4 | 170.79 | 78.11 | 4900.42 | 599.25 | 942.74 | 7465.77 | Sodium sulphate |
| 5 | 932.95 | 238.04 | 17,400.86 | 426.39 | 661.24 | 27,075.86 | Calcium chloride |
| 6 | 906.18 | 275.24 | 16,042.01 | 1807.12 | 539.09 | 24,056.37 | Sodium sulphate |
| 7 | 912.55 | 286.40 | 18,114.96 | 199.20 | 462.07 | 27,971.75 | Calcium chloride |
| 8 | 274.86 | 122.36 | 8987.88 | 436.92 | 647.96 | 15,130.63 | Calcium chloride |
| 9 | 1164.91 | 409.14 | 30,638.20 | 177.30 | 374.44 | 48,411.37 | Calcium chloride |

### 4.2. Macro-Morphology Analysis of the Hanging Rings

After being soaked in the pipelines of different positions for 3 months, the hanging rings were taken out, placed into the numbered sample bags in sequence, and taken to the laboratory for further analysis. From external observation (see Figure 2), most of the hanging rings were corroded with obvious corrosion characteristics, which could be divided into the following three categories:

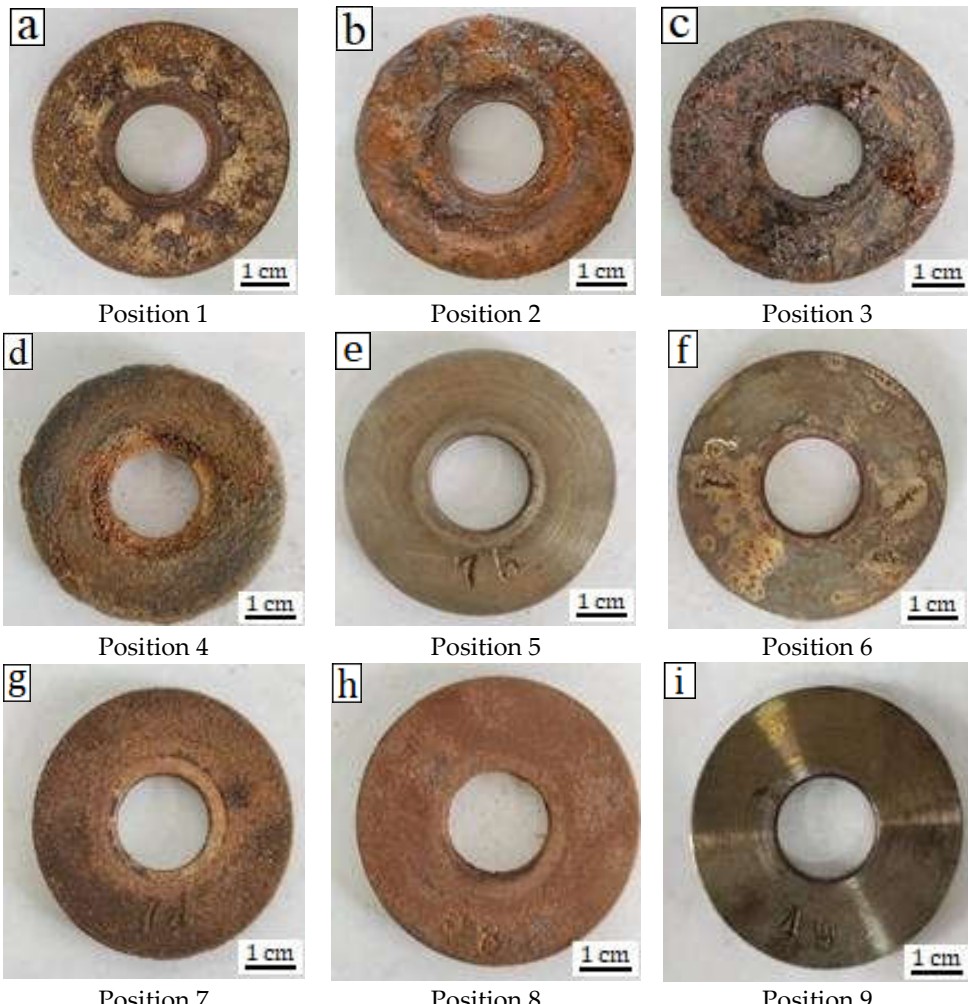

**Figure 2.** Macro-morphology of corrosion hanging ring at each site.

**Type I** Serious corrosion and scaling: At Positions #1, #2, #3, and #4, the corrosion and scaling of the hanging rings were very serious and could be observed even with the naked eyes, while either the corrosion/rust layer or scaling layer was found to be thick and uniformly distributed. In addition, the corrosion/rust and scaling layers had obvious delamination and there was falling off, indicating that either the corrosion/rust layer or scaling layer was not dense and loose enough, and the bonding force was small.

**Type II** Slight corrosion and scaling: At Positions #6, #7, and #8, the corrosion and scaling of the hanging rings were non-severe. The corrosion/rust layer or scaling layer was thin, and obvious pits could be seen on the surface.

**Type III** Basically free of corrosion and scaling: The corrosion degree of Positions #5 and #9 were not obvious, and the visible surface of Position #9 had an obvious metallic luster.

### 4.3. Micro-Morphology Analysis of the Hanging Rings

#### 4.3.1. Type I

Figure 3a and Table 3 show the SEM micro-morphology analysis and the EDS analysis results of the hanging ring at Position #1. Combined with macro-observations, it can be seen that the corrosion product layer was thick, the microstructure was loose and porous, and the structure was not dense. There were a large number of cracks and holes on the surface, which meant the corrosion medium was easy to penetrate from the surface of the scale layer. The EDS surface scanning results showed that the main elements on the surface were Ba, Sr, S, Fe, C, and others. The surface product layer was mainly not only a scaling one, but also

a corrosion product layer, which was a mixture layer mainly composed of $Ba_3Sr(SO_4)_4$ and a small amount of $FeCO_3$. From the above analyses, it could be concluded that corrosion at Position #1 was mainly induced by $CO_2$, including the accelerated corrosion under the scale caused by serious scaling.

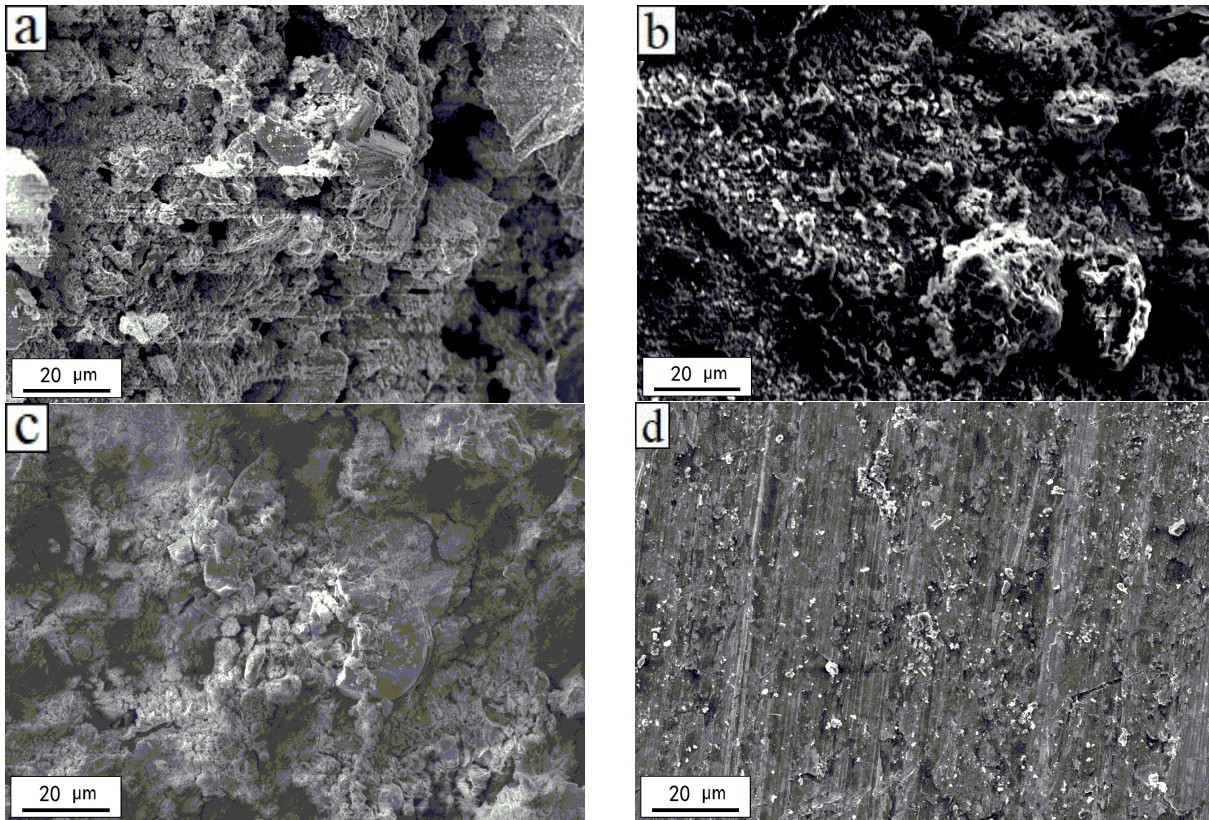

**Figure 3.** Microscopic morphology analysis of the hanging ring in (**a**) Position #1, (**b**) Position #4, (**c**) Position #8, and (**d**) Position #5.

**Table 3.** EDS surface scanning analysis results for the hanging rings at different positions.

| Position | Element | Weight Percentage | Atomic Percentage |
|---|---|---|---|
| #1 | C | 37.02 | 45.93 |
| | O | 56.08 | 52.24 |
| | S | 2.37 | 1.10 |
| | Ca | 0.30 | 0.11 |
| | Sr | 2.61 | 0.44 |
| | Ba | 1.62 | 0.18 |
| #4 | C | 7.99 | 11.97 |
| | O | 73.09 | 82.17 |
| | S | 5.60 | 3.14 |
| | Cl | 1.19 | 0.61 |
| | Sr | 7.06 | 1.45 |
| | Ba | 5.06 | 0.66 |
| #5 | C | 21.73 | 27.39 |
| | O | 76.12 | 72.02 |
| | Fe | 2.15 | 0.58 |
| #8 | C | 17.97 | 23.12 |
| | O | 78.65 | 75.95 |
| | Fe | 3.38 | 0.94 |

Figure 3b and Table 3 present the SEM micro-morphology analysis and the corresponding EDS surface scanning results of the hanging ring at Position #4. It can be seen that the micro-morphology of the corrosion product layer had a few cracks, the surface structure was dense, and there were blocky stacks in some parts. The EDS test results showed that the main elements on the surface were C, O, Cl, S, Ba, Sr, and others, and the main components were $BaSr(SO_4)_2$, and $CaCO_{32}$. Furthermore, the surface product layer was a mixture of corrosion products and scale layer. This phenomenon proved that $CO_2$, $H_2S$, and $Cl^-$ co-existed, and the corrosion and scaling were relatively serious.

### 4.3.2. Type II

Figure 3c and Table 3 show the micro-morphology analysis and the EDS analysis results of the hanging ring at Position #8. It can be seen that the surface corrosion products were less, the surface products were stacked in blocks, and the micro cracks were less. The EDS test results showed that the main elements on the surface were C, O, Fe, and others, and the main component was $FeCO_3$. The surface product layer was induced by minor $CO_2$ corrosion.

### 4.3.3. Type III

Figure 3d and Table 3 illustrate the micro-morphology analysis as well as the EDS analysis results of the hanging ring at Position #5. It can be seen that there was a small amount of corrosion products on the surface of the product layer. The processing traces were clearly visible, and the irregular shape of the surface products were scattered on the surface. The EDS test results showed that the main elements on the surface were C, O, Fe, and others, and the main component was $FeCO_3$. The surface product layer was a corrosion product with $CO_2$ corrosion, and such corrosion was minor.

### *4.4. Compositional Analysis of the Corrosion Product Layer*

For some hanging rings with severe corrosion, the surface product layer was thick. We scraped this product layer off with a blade, then dried and ground it to powder with an agate hammer and surface dish, followed by XRD chemical composition analysis. The compositions of the product layers with the XRD analysis of the hanging rings are illustrated in Figure 4. It can be seen from the figure that the composition of the product layer of Position #3 was mainly $BaSr_3(SO_4)_4$, $FeCO_3$, and small amounts of FeS; those at Positions #6 and Position #8 were mainly composed of $BaSr(SO_4)_2$ and $CaCO_3$, and that at Position #9 was mainly composed of FeS, $BaSr(SO_4)_2$, and $CaCO_3$.

Combined with the previous EDS test results, it can be seen that the product at Position #3 was mainly $Ba_3Sr(SO_4)_4$ scale due to scaling, but the scale surface was loose. Since there were many holes and cracks on this product layer, it could not play a protective role, resulting in accelerated corrosion under the scale. Since most of the $CO_2$ corrosion damage was pitting corrosion at the damaged film, it could be judged that the corrosion at Position #3 was mainly $CO_2$ corrosion, and a certain amount of $H_2S$ accelerated the corrosion process. Position #6 had a serious scaling phenomenon, and the scale layer was not dense. $Cl^-$ and other corrosive ions could easily penetrate through the cracks, resulting in accelerated pitting corrosion under the scale. The corrosion and scaling mechanisms of Position #8 were mainly scaling accompanied by $CO_2$ corrosion, and the synergistic effect of corrosion and scaling accelerated the occurrence of corrosion and scaling. The coexistence of $H_2S$, $CO_2$, and $Cl^-$ corrosion existed at Position #9, and the content of FeS in the product was high so that the $H_2S$ corrosion was serious. Since a certain amount of $H_2S$ can inhibit corrosion, the hanging ring of Position #9 had onlyslight corrosion and an obvious luster on the surface.

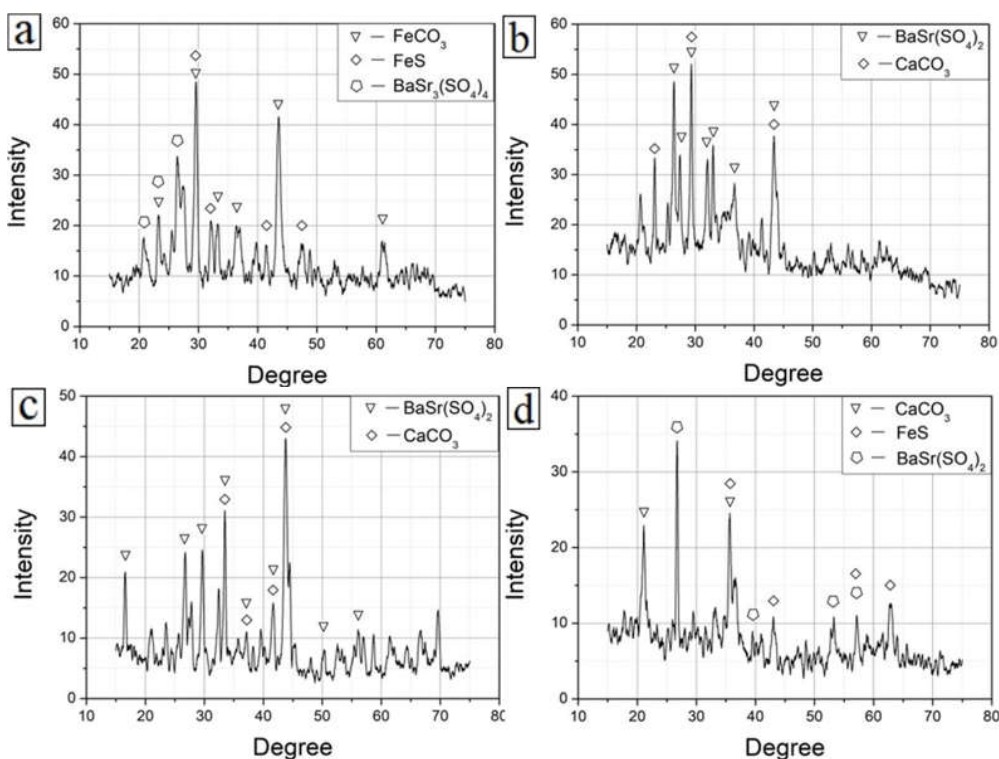

**Figure 4.** The XRD analysis results for hanging rings at: (**a**) Position #3, (**b**) Position #6, (**c**) Position #8, and (**d**) Position #9.

*4.5. Evaluation of Corrosion Inhibitors*

4.5.1. Open Circuit Potential Tests

Table 4 shows the open circuit potential test results of the blank sample and five corrosion inhibitors with different concentrations. It can be seen that the potential of the five inhibitors changed after adding different concentrations of inhibitors, indicating that the inhibitors had certain effects, but the underlying mechanisms were different.

**Table 4.** Open circuit potential test results of corrosion inhibitor.

| Open Circuit Potential/mV / Concentration/ppm | HHS08-1# | HHS08-3# | PD43-1# | RX-21# | YD-03# |
|---|---|---|---|---|---|
| 0 | | | −654.903 | | |
| 50 | −683.579 | −635.609 | −591.592 | −584.503 | −676.496 |
| 80 | −609.474 | −632.350 | −576.529 | −572.260 | −665.889 |
| 100 | −646.365 | −611.680 | −586.034 | −618.911 | −644.755 |
| 120 | −673.752 | −577.737 | −596.989 | −635.958 | −671.883 |

The open circuit potential of the HHS08-1# inhibitor was the highest at 80 ppm, which was −609.474 mV. With an increase in the corrosion inhibitor concentration, such an open circuit potential first increased and then decreased. The HHS08-3# corrosion inhibitor had the highest open circuit potential of −577.737 mV at 1200 ppm with a monotonic increase with increase of the corrosion inhibitor concentration. At a concentration of 80 ppm for the PD43-1# and RX-21# corrosion inhibitors, the open circuit potential, respectively, reached a maximum of −576.529 and −572.260 mV, first increasing and then decreasing as the inhibitor concentration increased. Finally, the open circuit potential of the YD-03# corrosion inhibitor had the highest value of −644.755 mV at 100 ppm, while it first increased and then decreased with an increase in the corrosion inhibitor concentration.

According to corrosion kinetics, the open circuit potential represents the corrosion tendency of a sample. It can be seen that, compared with the blank sample without corrosion

inhibitor (open circuit potential: −654.903 mV), the maximum corrosion potential increased after the addition of five corrosion inhibitors, indicating that the corrosion inhibitor played a certain role in reducing the corrosion tendency of the sample. Among them, the open circuit potentials of the HHS08-3#, RX-21#, and PD43-1# increased obviously with better slow-release effects.

### 4.5.2. Polarization Curve Tests

Figure 5a shows the polarization curves of the blank sample and the HHS08-1# inhibitor at different concentrations of 50, 80, 100, and 120 ppm. It can be seen that, after adding the corrosion inhibitor, the self-corrosion potential increased at different concentrations of the corrosion inhibitor, and the self-corrosion potential of 50 ppm increased the most with a value of −773.300 mV, which was significantly higher than that of the blank sample (i.e., −933.340 mV). The self-corrosion current density showed that the polarization curve did not shift to the left obviously, and the self-corrosion current density was close to that of the blank sample with a minor change.

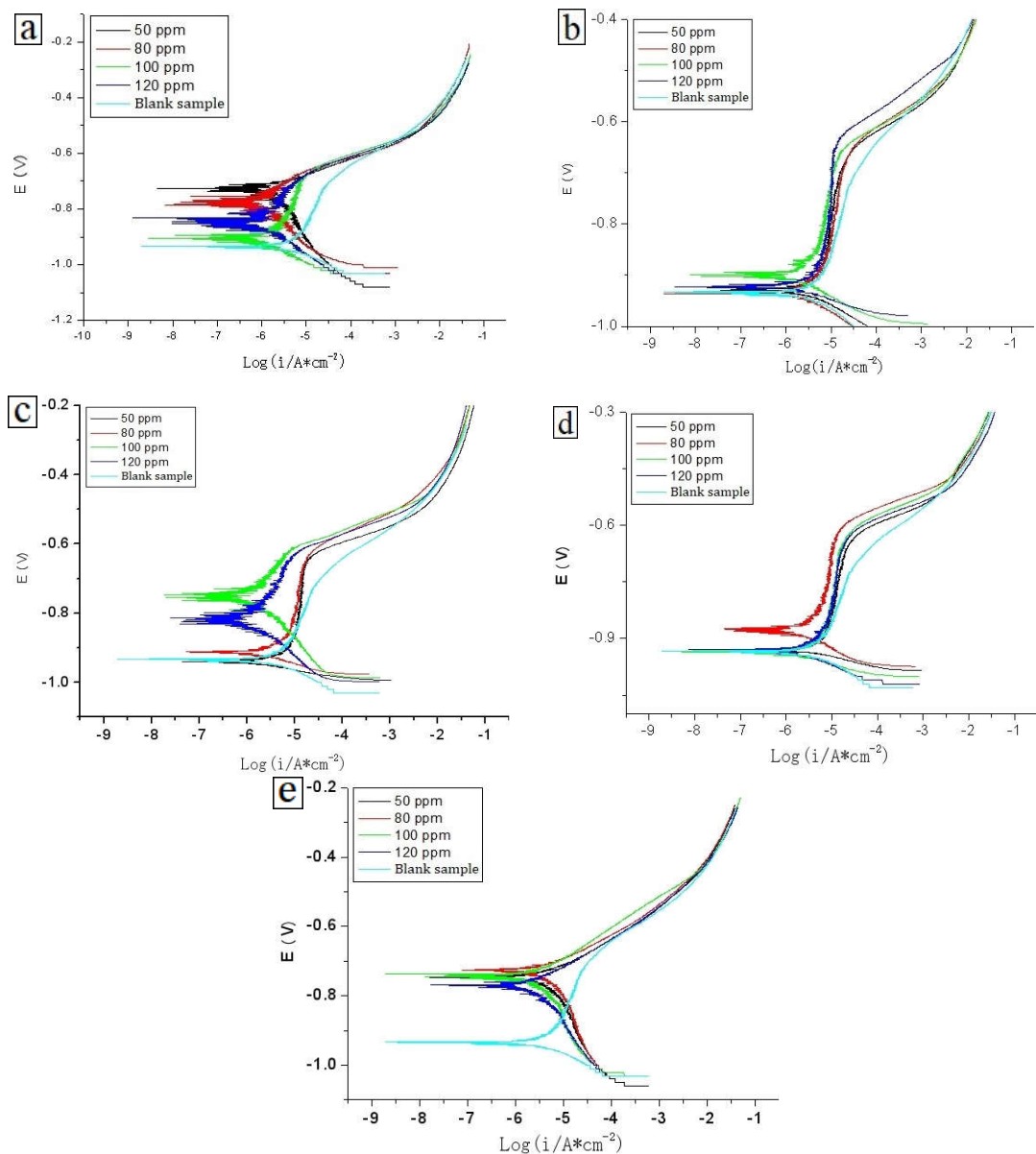

**Figure 5.** Polarization curves of the corrosion inhibitors: (**a**) HHS08−1#, (**b**) HHS08−3#, (**c**) PD43−1#, (**d**) RX−21#, and (**e**) YD−03# with different concentrations.

Figure 5b shows the polarization curve of the blank sample and the HHS08-3# inhibitor at different concentrations of 50, 80, 100, and 120 ppm. It can be seen that, after adding the corrosion inhibitor, the self-corrosion potential increased under different concentrations of the corrosion inhibitor. In particular, the self-corrosion potential at 100 ppm increased the most to −895.252 mV, which was higher than the self-corrosion potential (−933.340 mV) of the blank sample, but the improvement was limited. As can be seen from the self-corrosion current density, the polarization curve moved to the left at the concentration of 100 ppm, but this was not obvious. The self-corrosion current density was close to that of the blank sample with a minor change.

Figure 5c shows the polarization curve of the blank sample and PD43-1 inhibitor with different concentrations of 50, 80, 100, and 120 ppm. It can be seen that, after adding the inhibitor, the self-corrosion potential of the inhibitor increased at all concentrations except the concentration of 50 ppm. Especially notable was the fact that the self-corrosion potential of 100 ppm increased the most with its maximum of −754.184 mV, which was significantly higher than that of the blank sample (−933.340 mV). However, it can be seen from the self-corrosion current density that the polarization curve of the inhibitor moved to the left at all concentrations except the concentration of 50 ppm, and the self-corrosion current density moved to the left most obviously at the concentration of 100 ppm, falling to $1.22 \times 10^{-6}$ A/cm$^2$. According to the principle of corrosion thermodynamics, this showed that the optimal concentration of the PD43-1 inhibitor was 100 ppm, at which the sample had a better corrosion protection effect with an increase in both the self-corrosion potential and corrosion resistance. Furthermore, the self-corrosion current density was significantly reduced, indicating that the corrosion rate significantly reduced and, thus, at this concentration, the corrosion resistance of the sample significantly improved.

Figure 5d shows the polarization curve of the blank sample and the RX-21# inhibitor with different concentrations of 50, 80, 100, and 120 ppm. It can be seen that, after adding the corrosion inhibitor, the self-corrosion potential increased under different concentrations of the corrosion inhibitor. The self-corrosion potential at 80 ppm increased the most with its value of −917.577 mV, which was slightly higher than that of the blank sample (−933.340 mV). The self-corrosion current density showed that the polarization curve did not shift to the left obviously, and the self-corrosion current density was close to that of the blank sample with a minor change.

Figure 5e shows the polarization curve of the blank sample and the YD-03# inhibitor with different concentrations of 50, 80, 100, and 120 ppm. It can be seen that, after adding the corrosion inhibitor, the self-corrosion potential increased significantly under different concentrations of corrosion inhibitor with its maximum of −726.676 mV at 80 ppm, which was significantly higher than that of the blank sample (−933.340 mV). According to the self-corrosion current density, it can be seen that the polarization curve shifted to the left, especially at the concentration of 100 ppm, at which the self-corrosion current density dropped to $5.16 \times 10^{-6}$ A/cm$^2$.

According to the open circuit potential and polarization curve test results of the blank sample and the five in-service inhibitors, it can be seen that the five in-service inhibitors had certain corrosion inhibition effects, but the effects were different. The two better inhibitors were PD43-1# inhibitor and YD-03# inhibitor, respectively. After these two inhibitors were injected, the self-corrosion potential of the blank sample increased significantly but the self-corrosion current density decreased significantly. In addition, the optimal injection concentration of the PD43-1# corrosion inhibitor was found to be 100 ppm, at which the self-corrosion potential of the sample increased significantly to −754.184 mV and the self-corrosion current density decreased greatly to $1.22 \times 10^{-6}$ A/cm$^2$.

The optimum concentration of the YD-03# corrosion inhibitor was 80 ppm, at which the self-corrosion potential of the sample increased dramatically to −726.676 mV and the self-corrosion current density decreased significantly to $5.16 \times 10^{-6}$ A/cm$^2$. It can be seen that the corrosion electrochemical performance of the samples significantly improved at the optimal concentrations of the two inhibitors.

## 5. Discussion

For pure $CO_2$ corrosion, $CO_2$ gas reacting with water can produce $H_2CO_3$, resulting in a series of chemical reactions [26], which are shown in the following formulae:

$$CO_2 + H_2O \rightarrow H_2CO_3 \tag{1}$$

$$H_2CO_3 \rightarrow HCO_3^- + H^+ \tag{2}$$

$$HCO_3^- \rightarrow CO_3^{2-} + H^+ \tag{3}$$

Anodic reaction:

$$Fe \rightarrow Fe^{2+} + 2e^- \tag{4}$$

Anode product reaction:

$$Fe^{2+} + CO_3^{2-} \rightarrow FeCO_3 \tag{5}$$

For pure $H_2S$ corrosion, since $H_2S$ is extremely soluble in water, a large number of hydrogen ions are ionized in water [27], and its reaction in water is shown in the following formule:

$$H_2S \rightarrow H^+ + HS^- \tag{6}$$

$$HS^- \rightarrow H^+ + S^{2-} \tag{7}$$

Anodic reaction:

$$Fe \rightarrow Fe^{2+} + 2e^- \tag{8}$$

Anode product reaction:

$$Fe^{2+} + S^{2-} \rightarrow FeS \tag{9}$$

In an environment with both $CO_2$ and $H_2S$, when the $CO_2$ concentration is high and the $H_2S$ concentration is low, $CO_2$ corrosion is dominant. When the concentration of $CO_2$ is low and the concentration of $H_2S$ is high, it is mainly $H_2S$ corrosion [28]. In these experiments, the corrosion product generated in the coexistence environment of $CO_2$ and $H_2S$ was mainly FeS, indicating that the corrosion mechanism was mainly $H_2S$ corrosion. The corrosion of carbon steel by $H_2S$ is mainly realized by generating corrosive ion $H^+$, which reacts to generate H atoms and attaches to the matrix, causing corrosion to carbon steel [29]. At the same time, $CO_2$ dissolves in water to form $H_2CO_3$, and then $H_2CO_3$ decomposes to form $H^+$, which further increases the concentration of $H^+$ and corrodes the steel matrix. Therefore, the hanging ring at Position #4 showed serious corrosion behavior. The synergistic corrosion effect of $CO_2$ and $H_2S$ could be attributed to $CO_2$ enhanced $H_2S$ corrosion in essence, which was consistent with the findings documented elsewhere [29].

The water ion analysis results showed that the water sample contained a large amount of $Cl^-$, and such a high concentration of $Cl^-$ causes local acidification and pitting corrosion at the iron oxide/solution interface. This can be attributed to the autocatalytic acidification mechanism of chloride ions [30], and the reaction process is listed as follows:

$$Fe \rightarrow Fe^{2+} + 2e^- \tag{10}$$

$$Fe^{2+} + 2Cl^- \rightarrow FeCl_2 \tag{11}$$

$$FeCl_2 + 2H_2O \rightarrow Fe(OH)_2 + 2Cl^- \tag{12}$$

$$6FeCl_2 + O_2 + 6H_2O \rightarrow Fe_3O_4 + 12H^+ + 12Cl^- \tag{13}$$

As the concentration of metal cations in the hole of a handing ring increases, chloride ions migrate to maintain electrical neutrality so that a concentrated solution of metal chloride is formed in the hole, which can continue to maintain the activated state of the metal surface in the hole. As a result of chloride hydrolysis, the acidity of the medium in the hole of a hanging ring increases, which accelerates the dissolution of the anode, further

developing the corrosion hole, and forming a closed cell at the hole. After the closed cell is formed, the hydrolysis of chloride further increases the acidity of the medium. The increase of acidity further accelerates the dissolution rate of the anode and accelerates the high rate of the corrosion hole, which can erode through the metal section. In addition, scaling and corrosion promote and cooperate with each other. At present, the scale types of the Jiyuan pool are mainly $BaSr(SO_4)_2$ and $CaCO_3$, which account for more than 90%. At the same time, the content of $H_2S$ and $Cl^-$ in individual production layers is high, which promotes the corrosion of surface pipelines. Previous studies have also confirmed this finding [31,32].

## 6. Conclusions

In this paper, corrosion hanging rings were used in field experiments. The micromorphologies of the corrosion hanging rings were observed by using SEM, and the surface composition of the corrosion hanging rings analyzed with XRD. The main conclusions are summarized as follows:

The results of water ion analysis showed that the water ions of each position in the Jiyuan pool were rich in $Ca^{2+}$, $Ba^{2+}$, $SO_4^{2-}$, and $HCO_3^-$, and barium scale and calcium carbonate scale easily formed.

The results of corroded hanging rings showed that the $CO_2$ corrosion mechanism is common in the Jiyuan pool. At some severe corrosion positions, $CO_2$ and $H_2S$ corrosion coexisted, and a high concentration of $Cl^-$ was also an important cause of pitting corrosion.

The $CO_2/H_2S$ synergistic corrosion mechanism dominated by $H_2S$ can be attributed to $CO_2$ enhanced $H_2S$ corrosion, in essence.

It is suggested to optimize the optimization and injection of corrosion inhibitor in the Jiyuan oilfield. To improve the anti-corrosion process in the pipeline, using internal anti-corrosion coating, as well as introducing new technologies to improve water quality and reduce corrosive ions, such as the polarization purification method, could be technically viable.

**Author Contributions:** Writing—review and editing, Y.X.; writing—original draft preparation, Y.L.; methodology, Y.Y. and D.Y.; investigation, Q.G., J.J. and S.L. (Shubin Lei); data curation, Y.W., L.W. (Lei Wang) and L.W. (Lei Wen); conceptualization, S.L. (Shilei Li). All authors have read and agreed to the published version of the manuscript.

**Funding:** The present work has been financially supported by National Natural Science Foundation of China Project (Grant No.: 12102340), Young Scientific Research and Innovation Team of the Xi'an Shiyou University (Grant No.: 2019QNKYCXTD14), State Key Laboratory of Metastable Materials Science and Technology (Grant No.: 202111), State Key Lab of Advanced Metals and Materials (Grant No.: 2021-Z06), Opening project fund of Materials Service Safety Assessment Facilities (Grant No.: MSAF-2021-101), China Scholarship Council Foundation (Grant No.: 202208615046) and Postgraduate Innovation and Practical Ability Training Program of the Xi'an Shiyou University (Grant No.: YCS21211056).

**Institutional Review Board Statement:** Not applicable.

**Informed Consent Statement:** Not applicable.

**Data Availability Statement:** Not applicable.

**Acknowledgments:** The authors acknowledge a Discovery Grant and a Collaborative Research and Development (CRD) Grant from the Natural Sciences and Engineering Research Council (NSERC) of Canada to D. Yang.

**Conflicts of Interest:** The authors declare no conflict of interest.

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
