# Peer review of "Identification and Analysis of Corrosion Mechanisms for Ground Pipelines with Hanging Rings"

_coatings, doi:10.3390/coatings12091257_

Round 1

Author Response

Editor of Coatings. Electronic submission Xi′an, August 9th 2022 Dear Editor: Please find enclosed the revised manuscript entitled “Identification and Analysis of Corrosion Mechanisms for Ground Pipelines with Hanging Rings” with reference coatings-1840829_R1. The changes with respect the former text are in yellow colour. Moreover, in the next pages of this letter the “response to reviewers” is included. We hope that you find the paper suitable for publication in your Journal. Looking forward to hearing from you soon, Yours sincerely Dr. Wang Lei School of Materials Science and Engineering Xian Shiyou University, Xi′an, shaanxi, 710065, PR China Tel. Phone: (86)-15771921910 E-mail address: [email protected], [email protected]

Reviewer(s)' Comments to Author: The paper looks interesting from the viewpoint of the Identification and Analysis of Corrosion Mechanisms for Ground Pipelines with Hanging Rings. I suggest that it could be accepted after major revision, and here are my suggestions for the authors.

Introduction: The corrosion mechanisms on pipeline are missing.

Yes, we have added the corrosion mechanism of pipelines in the "Introduction" section.

Introduction: The corrosion effect of different ions on ground pipelines must be added.

Yes, we have explained the corrosion effect of different ions in the manuscript.

Introduction: The electrochemical performance of common and uncommon corrosion inhibitors on pipeline are missing.

Yes, we have added the introduction of common corrosion inhibitors.

Introduction: The research goal is not clear.

We rephrased the research objectives to make them clearer.

Introduction: You should state clearly which is the problem or the question that you are trying to solve.

Yes, we have more clearly stated the problems to be studied.

Experimental: It is important to show the chemical composition of the hanging ring used in the experiment and the author must mention how the chemical composition was obtained.

We have added the chemical composition of the investigated hanging ring and explained its test method.

Experimental: The author must explain better why the hangings rings were placed at different stations along the pipeline and removed after 3 months.

Yes, we have explained in the manuscript.

Experimental: Explain better the main of serious corrosion (Line 106).

We have explained the possible causes of serious corrosion in the “experiment” part.

Experimental: Why were collected water samples of 9 different positions?

It is mainly due to the experimental design. Considering the production data of the oil field in the early stage, the positions where corrosion perforation may occur in the pipeline are found and monitored.

Experimental: Why was used 50, 80, 200 and 120 ppm of the filling concentration or each inhibitor?

This is the conventional addition dose of corrosion inhibitor in the laboratory test. At the same time, it has been considered that there are certain differences in the addition dose.

Results: Why the author mentions that the corrosion product layer is a mixture layer mainly composed of Ba3Sr(SO4) and a small amount of FeCO3 and the corrosion is mainly induced by CO2?

(See section 4.3, for Figure 3a and Table 2) This product layer includes a mixture of corrosion products and scaling products. Among them, FeCO3 is the product caused by carbon dioxide corrosion, while Ba3Sr(SO4) is the product formed by scaling, which also causes corrosion under the scale.

Results: The presence of Fe is not shown by EDS analysis in Table 2 for Figure 3b. For that, the FeS, FeCl2 components cannot be formed.

Yes, we have made modification in the manuscript.

Results: Why the XRD analysis shows results of different samples compared with SEM and EDS analysis? I suggest comparing the same samples to correlate the results.

The XRD test is mainly for the samples with thick product layer, while the SEM / EDS test is for the micro morphology of the hanging ring, so the selected samples are different.

Results: Which samples were used to carry out boot the open circuit potential and polarization curve test results?

The evaluation of corrosion inhibitor was carried out on 20 # steel, which can reflect the corrosion inhibition effect of corrosion inhibitor alone.

Results: You cite a lot of other contributions, and it is not clear what is what you found in this paper, not found before or different from before.

Compared with many previous works, most of them used indoor experiments, the main difference of this work is that the research method used is field experiments, and on this basis, the mechanism of scaling and corrosion is analyzed.

Results: The author must add the different standard PDF numbers (JCPDS) used to identify the different phases formed. The authors should properly align the objectives of the study in Abstract, Introduction, and in Conclusion parts.

Yes, we have made corresponding changes.

Results and discussion: I suggest presenting a better literature revision, even to highlight the results of this investigation with the literature.

Yes, we rearranged the references and compared some similarities and differences in the research work.

Results: Is not clear what is what you found in this paper, not found before, or different from Before.

This work is mainly based on the on-site hanging ring analysis of scaling and corrosion mechanism. Based on the characterization of water quality ions and corrosion scaling products, the cause of formation is analyzed, and the corresponding slow-release agents are evaluated.

Results: You don´t need to make everything but try to compare your results with others published early and discuss the differences if there are Conclusion: It must be improved.

Yes, we rearranged the references and compared some similarities and differences in the research work.

Reviewer 2 Report

Manuscript: Identification and Analysis of Corrosion Mechanisms for Ground Pipe-lines with Hanging Rings

In general, the article is interesting, and the information provided by the authors is important.

After the review of the manuscript, I have the following comments.

1). In section 4.3.1 (lines 160, 161), the authors indicate, “Combined with macro-observations, it can be seen that the corrosion product layer is thick” It could be important to describe compared with what the corrosion product layer is thick. Probably cross-section microphotography of all the corroded surfaces could better explain the affirmation.

Author Response

Reviewer(s)' Comments to Author:

In general, the article is interesting, and the information provided by the authors is important.

After the review of the manuscript, I have the following comments.

1). In section 4.3.1 (lines 160, 161), the authors indicate, “Combined with macro-observations, it can be seen that the corrosion product layer is thick” It could be important to describe compared with what the corrosion product layer is thick. Probably cross-section microphotography of all the corroded surfaces could better explain the affirmation.

In this experiment, the macro morphology of corrosion products (Fig. 2) can better reflect the formation of corrosion products, so the cross-sectional micrograph is not given.

Reviewer 3 Report

1.The scale bar must be introduced in Fig. 1 and Fig. 2

in order to reveal the exact size of the sample.

2. Instead of Table 1 and Table 2, appropriate graphical

sketches would make the presentation better.

Author Response

Reviewer(s)' Comments to Author:

1.The scale bar must be introduced in Fig. 1 and Fig. 2, in order to reveal the exact size of the sample.

Yes, we have added rulers to these figures.

2. Instead of Table 1 and Table 2, appropriate graphicalsketches would make the presentation better.

As the data in Table 1 and table 2 are complex, and in order to enrich the diversity of data expression, these data are expressed in the form of tables.

Round 2

Reviewer 1 Report

My comments were attended by the authors